

# 1064 nm Raman lidar for extinction and lidar ratio profiling: Cirrus case study

Moritz Haarig[1], Ronny Engelmann[1], Albert Ansmann[1], Igor Veselovskii[2], David N. Whiteman[3], and Dietrich Althausen[1]

[1]Leibniz Institute for Tropospheric Research, Leipzig, Germany
[2]Physics Instrumentation Center, Moscow, Russia
[3]NASA, GSFC, Greenbelt, Maryland, USA

*Correspondence to:* M. Haarig
(haarig@tropos.de)

**Abstract.** For the first time, vertical profiles of the 1064 nm particle extinction coefficient obtained from Raman lidar observations at 1058 nm (nitrogen rotational Raman backscatter) are presented. We applied the new technique in the framework of test measurements and performed several cirrus observations of particle backscatter and extinction coefficients, and corresponding extinction-to-backscatter ratios at the wavelengths of 355, 532m and 1064 nm.

## 1 Introduction

The Raman lidar technique is used to measure particle backscatter and extinction coefficients, and the extinction-to-backscatter ratio (lidar ratio) since more the 25 years (Ansmann et al., 1990, 1992). This technique opened a solid way for comprehensive studies of anthropogenic pollution, desert dust, and biomass burning smoke at several wavelengths (Mattis et al., 2002, 2004; Müller et al., 2005, 2007) and found broad application in the European Aerosol Research Lidar Network, EARLINET (Pappalardo et al., 2014). Particle extinction profiling is a basic requirement for a successful retrieval of microphysical aerosol properties by means of inversion methods (Müller et al., 1999; Veselovskii et al., 2002).

However, it remained an open issue throughout the years to measure the aerosol extinction coefficient at 1064 nm. State-of-the-art multiwavelength aerosol lidars use Nd:YAG lasers transmitting laser pulses at 355, 532 and 1064 nm. Techniques have been developed to determine height profiles of particle backscatter and extinction coefficients at 355 and 532 nm, but not at 1064 nm. Only the 1064 nm backscatter coefficient is delivered. The improved coverage of the wavelength spectrum by backscatter and extinction pairs at all three Nd:YAG wavelengths will permit an improved, more robust characterization of aerosols in terms of optical and microphysical properties with lidar.

Here, we report the first Raman lidar measurement of particle extinction at 1064 nm, conducted in October 2015 with a triple-wavelength Raman lidar. In this short article, we present a cirrus case. Cirrus provides almost optimum conditions for checking the performance of the new lidar approach. The overlap between the laser beam and the receiver field of view (RFOV) is complete in the far range of the lidar and does not introduce any bias in the retrieval products. Furthermore, the suppression of elastic backscatter light in the used 1058 nm Raman channel (anti-Stokes rotational Raman signal passing a filter centered at



1058 nm) can be checked at extremely large ice-crystal backscatter conditions. In cirrus, the particle backscatter, extinction, and lidar ratio profiles for all three wavelength (355, 532, and 1064 nm) should be rather similar because of wavelength-independent backscattering and extinction when the used laser wavelengths are small compared to the size of ice crystals.

In the Sect. 2, we provide details to the multiwavelength lidar and the data analysis methods. Sect. 3 discusses the cirrus
measurement. A few concluding remarks are given in Sect. 4.

## 2 Instrumentation

We implemented the new Raman channel in our containerized multi-wavelength polarization/Raman lidar BERTHA (Backscatter Extinction lidar-Ratio Temperature Humidity profiling Apparatus) (Althausen et al., 2000; Haarig et al., 2015) which has been used over the past 20 years in 12 major field campaigns in Europe, Asia, Africa and the Caribbean (Wandinger et al.,
2002; Ansmann et al., 2002; Franke et al., 2003; Tesche et al., 2009, 2011; Haarig et al., 2015). In 2012, BERTHA was redesigned to allow particle linear depolarization measurements at 355, 532, and 1064 nm, simultaneously. BERTHA permits us to measure particle optical properties by applying the Raman lidar method and, in addition, by means of the High Spectral Resolution Lidar (HSRL) technique at 532 nm (Althausen et al., 2012).

Two Nd:YAG lasers (Continuum, Powerlite) transmit linearly polarized laser pulses at 355 and 1064 nm (first laser) and
at 532 nm (second laser). Two additional linear polarizers clean the polarization of the outgoing light. The repetition rate is 30 Hz and the pulse length 50 ns. The pulse energies are 1000 mJ (1064 nm), 800 mJ (532 nm) and 120 mJ (355 nm). The beams are expanded tenfold and pointed almost vertically into the atmosphere at an off-zenith angle of 5°. The receiver field of view (RFOV) is 0.8 mrad. A 53 cm Cassegrain telescope collects the backscattered light. The signals are detected with a range resolution of 7.5 m and a time resolution of 10 s. To avoid overloading of the photomultipliers (PMTs) in the near range
we restricted the maximum count rate to 20 MHz for the signal maximum in about 500 m height.

A sketch of the receiver unit is given in Fig. 1. We replaced the third cross-polarized signal channel (for 1064 nm) by the 1058 nm rotational Raman channel. The design and realization of the 1058 nm interference filter used for 1064 nm backscatter and extinction profiling was motivated by the successful demonstration of high-quality aerosol measurements with an interference-filter-based 532 nm rotational Raman lidar by Veselovskii et al. (2015). The same methodological approach, for
the laser wavelength of 532.12 nm in Veselovskii et al. (2015), is applied here for the wavelength of 1064.24 nm. The interference filter is centered at about 1058 nm (Alluxa, Santa Rosa, CA, http://www.alluxa.com). The transmission band is from 1053-1062 nm with a transmission >90% in this wavelength range. For the laser wavelength of 1064.24 nm, the transmission is specified to be 0.005%. A broad band blocking by more than four orders magnitude from 350-1100 nm is realized.

When using a laser wavelength of 1064.24 nm, 88% of the total intensity of the rotational Raman backscatter spectrum (anti-
Stokes lines) can pass the 1053-1062 nm filter. The detected Raman backscatter intensity is only weakly temperature-dependent (4% increase of the measured rotational Raman signal for a temperature decrease from 300 K to 230 K). To guarantee no crosstalk contributions in the rotational Raman channel even in the case of extremely high elastic backscattering, e.g., caused by





specular reflection by almost horizontally aligned cirrus crystals and vertical laser beam pointing, we used two Raman filters in front of the PMT.

The data analysis is performed following the EARLINET data analysis protocol (Pappalardo et al., 2004). In the computation of Rayleigh backscattering and extinction contributions to the measured lidar signals, we used GDAS (Global Data Assimila-
tion System) height profiles of temperature and pressure of the National Weather Service's National Centers for Environmental Prediction (NCEP, NOAA's Air Resources Laboratory ARL, https://www.ready.noaa.gov/gdas1.php).

## 3   Results

First test measurements with the new Raman channel were performed on 3 October 2015. Fig. 2 presents an observation performed on 12 October with a well aligned lidar during a long-lasting cirrus event. The ice cloud layer showed a stable base
height at 6.5 km. The top height was detected at 10.5 km, probably coinciding with the tropopause. The backscatter intensity increased by two orders of magnitude at cloud base according to the 1064 nm elastic backscatter signal (right panel in Fig. 2). As mentioned, to avoid strong specular reflection effects the laser beams were pointing to an off-zenith angle of $5°$. The Raman signal profiles do not show any interference by strong elastic backscatter by ice crystals. Sufficient blocking of 1064 nm lidar returns was achieved by using two 1058 nm filters.

The 4 km deep, comparably homogeneous cirrus layers provided favorable conditions for a proper determination of the cirrus optical properties. The respective height profiles of particle backscatter and extinction coefficients, and lidar ratio at 1064 nm are shown in Fig. 3. Two hours of measurements are averaged. For comparison, the results for 355 and 532 nm are shown in addition. Similar results are expected at all three wavelengths because of wavelength-independent backscattering and extinction by ice crystals which are assumed to be at all very large ($> 100 \mu$m in diameter), at least in the lower half of the
cirrus layer where the coherent backscatter structures in Fig. 2 indicate fall streaks.

Keeping the shown uncertainty bars into consideration, similar profiles of the particle backscatter coefficients are obtained at all three wavelengths. All profiles are based on the analysis of the elastic-backscatter-to-Raman signal ratio profiles.

In the case of the extinction profiles, a good agreement between the 532 and 1064 nm extinction profiles (Raman lidar solutions) is visible. The 355 nm extinction coefficients are considerably smaller in the lower part of the cirrus layer. The
quality of the HSRL solutions will not be discussed here in detail. Not all necessary corrections (e.g., proper consideration of all cross-talk effects and near-range iodine-absorption saturation effects) are carefully applied in this example. However, in the lower part of the cirrus (7-8 km height range) with lowest level of signal noise and similar vertical window lengths in the regression analysis of the Raman signal profiles (2000-2500 m), the differences between the three extinction coefficients are comparably small. The cirrus extinction coefficients are 125 $\text{Mm}^{-1}$ (355 nm, 532 nm HSRL), 150 $\text{Mm}^{-1}$ (1064 nm), and
175 $\text{Mm}^{-1}$ (532 nm).

The main reason for the deviation of the 355 nm extinction coefficients from the ones at larger wavelengths is probably the use of a modeled temperature profile (GDAS model data) in the Raman lidar data analysis. Uncertainties in the Rayleigh backscatter and extinction corrections (caused by not well-known temperatures and vertical temperature gradients in the cirrus





layer) have the largest impact on the solutions for 355 nm. Molecular scattering and backscattering (including Raman backscattering) is higher by a factor of 5 and 81 at 355 nm, compared to 532 and 1064 nm, respectively. Another reason for the deviation of the 355 nm extinction profile could be multiple scattering (MS). This effect increases with decreasing wavelength. MS effects (forward scattered laser radiation remains in the RFOV so that the effective attenuation of laser radiation by ice crystal scattering is significantly reduced) leads to an apparent (effective) extinction coefficient and lidar ratio, about a factor of 1.5-2 lower than the respective single-scattering values. A third reason could be related to the cirrus crystal size distribution which caused stronger extinction at 532 and 1064 nm than at the shorter wavelength of 355 nm. However, accompanying Doppler lidar observations of crystal terminal velocity (about 0.7 m/s at cloud base) indicate large crystals (with diameters typically exceeding 300-400 $\mu$m) so that wavelength-independent backscattering and extinction is most likely.

The right panel in Fig. 3 shows for the first time, cirrus lidar ratios simultaneously measured at 355, 532, and 1064 nm. As expected, the lidar ratio is roughly equal in the ice cloud layer for the three wavelengths. After removing the MS impact from the derived effective lidar ratios of 22-28 sr (for the most trustworthy height range from 7-8 km height), the single-scattering related lidar ratios are in the range from 35-55 sr (for an MS factor of 1.5-2). Correspondingly, the apparent cirrus extinction coefficients are in the range fo 200-350 Mm$^{-1}$, and the cirrus optical thickness is about 0.45-0.6 after the correction of the MS influence.

## 4 Conclusions

For the first time, vertical profiles of the 1064 nm particle extinction coefficient obtained from Raman lidar observations around 1058 nm are presented. We applied the new technique to study cirrus backscatter and extinction coefficients, and corresponding cirrus lidar ratios at the wavelengths of 355, 532m and 1064 nm. A very powerful lidar was used for this study. The results are promising.

In the next step, Raman lidar observations at three wavelength should focus on aerosol layers (boundary layer, lower free troposphere) at very different conditions in terms of contribution by urban haze, biomass burning smoke, marine particles, and desert dust. Emphasis should also be put on the comparison with accompanying Aerosol Robotic Network (AERONET) photometer observations from 340-1640 nm as recently demonstrated by Veselovskii et al. (2016) by using a state-of-the-art dual-wavelength Raman lidar.

*Acknowledgements.* We thank Johannes Bühl for providing Doppler lidar observations of vertical velocity and estimated ice crystal sizes.





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

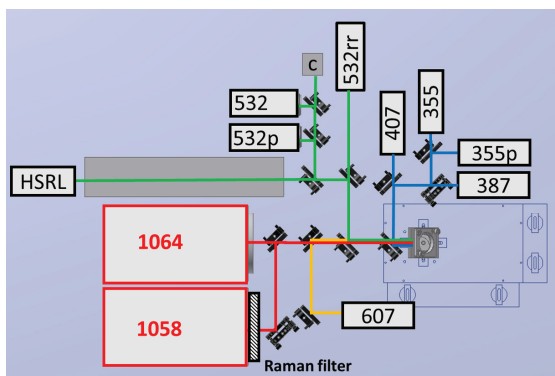

**Figure 1.** Sketch of the receiver unit of the lidar BERTHA with 13 detection channels (all operated in the photon counting mode): three elastic backscatter channels (355, 532 and 1064 nm), two cross-polarized elastic backscatter channels (355p, 532p), two nitrogen vibrational-rotational Raman channels (387 and 607 nm), three rotational Raman channels (532rr), one rotational Raman channel (1058), a water-vapor vibrational-rotational Raman channel at 407 nm and a HSRL channel at 532 nm. All detectors are photomultiplier tubes (PMT) from Hamamatsu (H10721P-110, except for 1058 and 1064 nm). For the 1058 and 1064 nm channels the PMTs R3236 from Hamamatsu are used. To reduce the signal noise, they are cooled down to lower than -30°C.



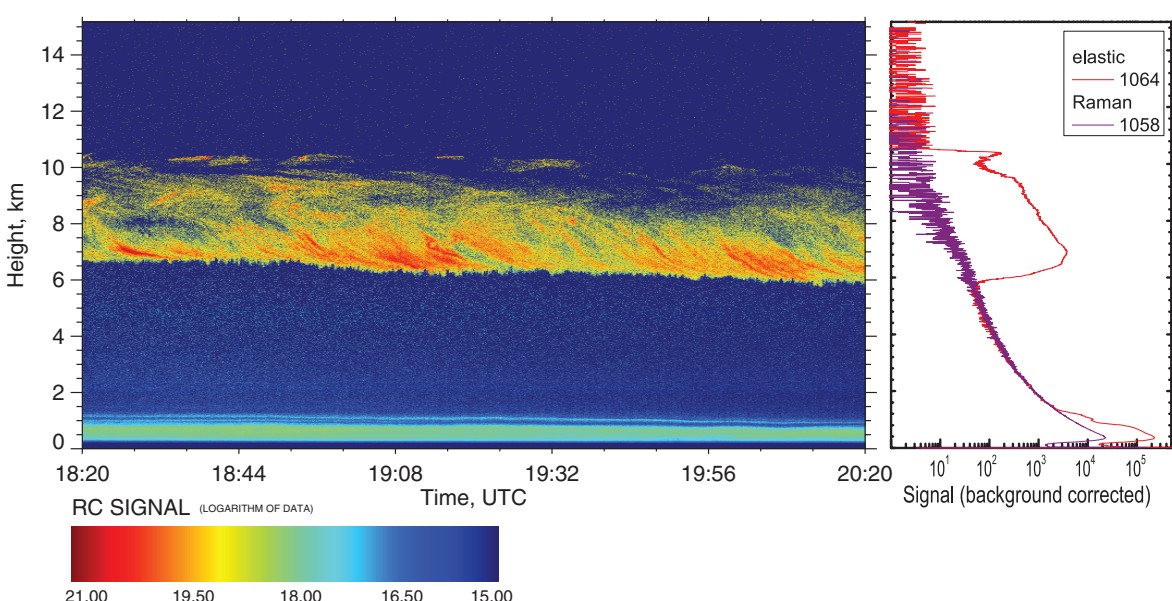

**Figure 2.** Cirrus observation at Leipzig, Germany, on 12 October, 2015, 18:20-20:20 UTC. The range-corrected 1064 nm lidar return signals indicate an ice cloud layer between 6.5 and 10.5 km height with virga in the lower part. In the right panel, the signal profiles (rotational Raman channel centered at around 1058 nm and 1064 nm elastic-backscatter channel) are averaged over the two-hour period shown in the left penal.



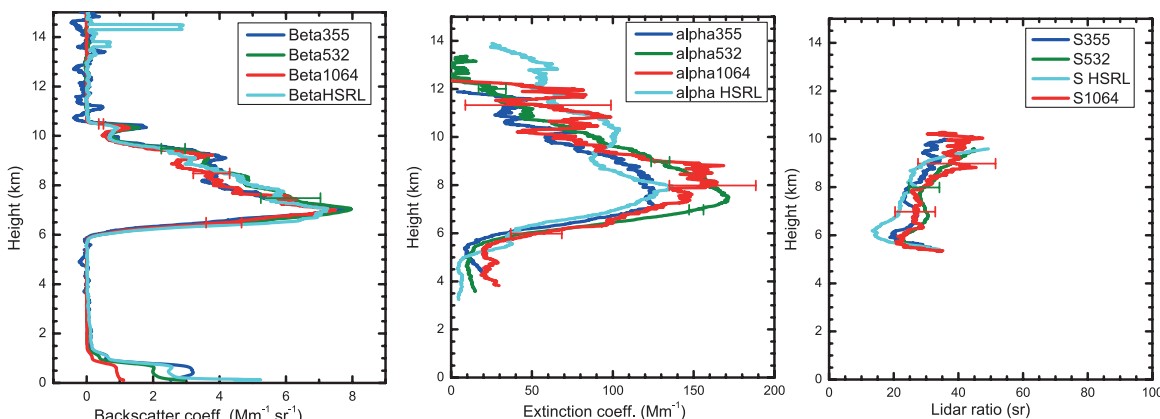

**Figure 3.** Particle backscatter coefficients (left), extinction coefficients (center), and corresponding lidar ratio (right) determined from the two-hour lidar measurement shown in Fig. 2 and taken at Leipzig on 12 October, 2015, 18:20-20:20 UTC. The Raman-lidar backscatter and extinction methods are used in the case of the 355, 532, and 1064 nm profiles. In addition HSRL solutions (532 nm) are shown. In the case of the backscatter profiles, a vertical smoothing length of 200 m is applied. The extinction profiles are obtained from a least-squares linear regression analysis applied to the 387, 607, and 1058 nm Raman signal profiles. The window lengths are 2000 m in the case of 355 nm and 532 nm, and 2500 m and 3200 m below and above 9 km height, respectively in the case of 1064 nm and the HSRL extinction profile. The lidar ratios are calculated for the same coarse height resolution. Error bars show the uncertainty caused by signal noise.