# Peer review of "1064 nm Raman lidar for extinction and lidar ratio profiling: Cirrus case study"

_Atmospheric Measurement Techniques, 2016_

## Short Comment (SC1) · 13 Apr 2016

Dear Authors,

I read your paper with great interest. The demonstrated ability to measure extinction directly at the Nd:YAG fundamental frequency is super and I was glad to see this. I agree that having extinction at this wavelength would be very useful for a range of aerosol research projects.

However, I have two questions that I hope that you can address in some manner:

1) Line 26: the interference filter is 9 nm wide. This would seem to suggest that these observations are only useful during the night (with this interference filter). Is that correct? What are the prospects of making extinction measurements at 1064 nm using

the Raman method in the daytime?

2) Signal-to-noise ratio will be an important limiter in this technique. If there is too little extinction over a layer (or the overall optical depth is too small), then it would be difficult to measure a good extinction profile. However, many aerosol layers have optical depths ranging from 0.1 to 0.25 (at 532 nm), but these layers are often closer to the lidar too. What sort of temporal averaging would be needed for these types of layers to get an accurate extinction profile?

Both of these comments are trying to understand how often Raman extinction measurements at 1064 nm would be able to be derived with useful error bars.

Thank you. This is good work, and I hope the editor will publish it.

---

## Referee Comment (RC1) · Anonymous Referee #1 · 10 May 2016

In their paper, M. Haarig et al. present a case of cirrus extinction profiling at 1064nm using a rotational Raman lidar. This is an very interesting paper demonstrating clear advancement in the field, and will be interesting to many lidar scientists. However, the paper needs improvements in terms of presentation, and in particular needs to provide more detailed information about the technique.

Specific comments

Introduction: The paper is missing a clear overview of aerosol extinction measurement techniques, and thus the novelty of what is presented. First, when discussing past measurements, the authors should make clear distinction of vibration and rotational Raman techniques. Rotational Raman should be in the title. Second, the presented method is an extension of Veselovskii et al. 2015 and this should be made clear in the

introduction.

Page 2, lines 29 - 30: Provide the source of these estimates (reference or your calculations).

Page 2, lines 31- 32: Provide some quantitative argument for your choice (why two and not three or one filters). What are the expected RR and elastic cross sections in a typical cirrus cloud?

Results: Collect in a paragraph all the scattered details about the lidar processing (e.g. vertical window lengths). Provide details and references of the multiple scattering corrections and which cases they are applied (and in which not). For clarity, provide also the basic equations for RR retrievals. Give detailed error estimation of the new product. What are the effects of polarization-dependent receiver transmission in these cirrus measurements(Mattis et al, Applied Optics 48(14):2742-51, 2009)

Page 3, lines 25 – 26: I don't see any value of the HSRL channel in this paper. Without all the necessary HSRL corrections, any agreement could be coincidental. I suggest to remove it from the final version of the paper.

Page 4, line 10-15: Compare your values with previously published values in the literature.

Figure 3. Change the line labels either to the letters "beta" and "alpha" or to the words "backscatter " and "extinction".

---

## Referee Comment (RC2) · Anonymous Referee #2 · 18 May 2016

The paper by Haarig at all presents for the first time extinction and lidar ratio measurements with a Raman lidar system simultaneously at 355 nm, 532 nm and 1064 nm. The newly applied measurements at 1064 nm are very valuable for retrieving optical and microphysical properties of clouds and Aerosols, and I am looking forward to see the first Raman measurements at 1064 nm for aerosol layers.

I recommend publication after answering some minor comments:

General questions:

What do you think is the required power of a lidar system to perform Raman measurements at 1064 nm?

Specific questions:

[Figure]

p. 3, l. 21: You should add the reference to Fig. 3 already here.

Figure 3: Error bars for measurements at 355 nm and HSRL not visible. Error bars including the retrieval uncertainties (e.g. caused by the use of modeled temperature profiles) should be added.

p. 3, l. 24: Are the smaller values at 355 nm significant if you consider all measurement and retrieval uncertainties?

p. 3, l. 25: If the HSRL measurement is not discussed, they can be skipped in this paper.

p. 4, l. 12: What do you mean by 'most trustworthy height range'?

p. 4, l. 13: Change 'range from' to 'range of'

p. 4, l 14: Change 'range fo' to 'range of'

---

## Author Comment (AC1) · 8 Aug 2016

**Dear Editor!**

We thank the three reviewers for careful reading and their suggestion to improve the paper. Before we go through all points of the reviewers, step by step, we want to give a brief overview of the main changes and improvements:

- We provide more information on the unique three wavelength cirrus measurement results in the abstract , we present the multiple-scattering-corrected optical properties.
- We improved the introduction and give an extended overview of the available techniques for routine aerosol profiling (HSRL, Raman lidar), and provide an extended list of references here. We give more details to the Raman lidar method, and introduce already in the introduction the basic idea developed by Igor Veselovskii.
- We describe the technique and the apparatus in more detail now in section 2.
- The result section (section 3) is carefully re-written. We carefully recalculated all Raman lidar
  profiles in the cirrus (checked scattering wavelength-dependence, temperature and pressure
  profile input on extinction and lidar ratio calculations). We now discuss in detail the influence of
  multiple scattering on the measured cirrus optical properties, compare the findings for the
  different wavelengths in more detail and provide an extended comparison with the available
  literature concerning multiple-scattering-corrected lidar ratios.
- Finally we introduced a new subsection 3.1 to discuss in an extended way, the potential of 1064 nm rotational Raman lidar to provide aerosol measurements in the lower troposphere with appropriate accuracy (10-20%) and resolution (one hour, 500 to 750 m vertical).

**Response to Anonymous Referee #1**

Introduction: The paper is missing a clear overview of aerosol extinction measurement techniques, and thus the novelty of what is presented. First, when discussing past measurements, the authors should make clear distinction of vibration and rotational Raman techniques. Rotational Raman should be in the title. Second, the presented method is an extension of Veselovskii et al. 2015 and this should be made clear in the introduction.

This is now improved (overview of aerosol extinction measurement techniques, only lidar can do it..., HSRL or Raman) as mentioned above. However, we do not see why we should provide a clear distinction between vibrational-rotation and pure rotational Raman lidar observations. We leave that out. To our opinion, the differences are minor, and the first attempts to use rotational Raman lidar for aerosol (extinction) profiling were made in the early 1980s (Arshinov and colleagues, and only published in ILRC proceedings, Aix en Provence, 1984).

Rotational Raman is added to the title.

Veselovskii et al. (2015) and the basic approach developed in this paper is already mentioned in the introduction (last paragraph).

Page 2, lines 29 - 30: Provide the source of these estimates (reference or your calculations).

There is no source (other than Igor Veselovskki...). All this information is based on computations of the co-author Igor Veselovskii.

*Page 2, lines 31-32: Provide some quantitative argument for your choice (why two and not three or one filters). What are the expected RR and elastic cross sections in a typical cirrus cloud?*

In Ansmann et al. 1992 (Appl. Phys. B), it is stated that the suppression of elastic backscatter in the Raman channels must be six orders of magnitude to avoid specular reflection effects (in the case of a zenith-pointing lidar). Now we use a lidar operated at an off-zenith angle of 5 degrees, so we need a supression by four orders of magnitude. By using two 1058 nm filters we obtain a suppression of almost five orders of magnitude ( $0.25 \times 10^{-4}$ ). All this isnow given and discussed in section2 (almost at the end, in the paragraph before the final paragraph).

Results: Collect in a paragraph all the scattered details about the lidar processing (e.g. vertical window lengths). Provide details and references of the multiple scattering corrections and which cases they are applied (and in which not). For clarity, provide also the basic equations for RR retrievals. Give detailed error estimation of the new product. What are the effects of polarization-dependent receiver transmission in these cirrus measurements (Mattis et al, Applied Optics 48(14):2742-51, 2009)

Yes, we do that now (collection of all scattered details about lidar data processing) in the first long paragraph of section 3.

However, we do not provide the basic equations of the Raman method, because this is done in very large detail in Veselovskii et al. (2015), we state that (first paragraph in section 2).

We do not discuss the uncertainties in the Raman retrievals, this has already been done many times during the last 20 years, and the most important error is caused by signal noise and this uncertainty is shown as error bars.

There are no polarization-dependent receiver transmission effects, we state that in section 2. We checked all this carefully.

Page 3, lines 25 – 26: I don't see any value of the HSRL channel in this paper. Without all the necessary HSRL corrections, any agreement could be coincidental. I suggest to remove it from the final version of the paper.

We leave the HSRL solutions in the figure. We provide the additional information that we will do a complete correction of all effects in the HSRL retrieval in a follow-up paper, which is inprepartion.

*Page 4, line 10-15: Compare your values with previously published values in the literature.*

**Done! ... see last paragraph in Section 3 (before section 3.1)**

*Figure 3. Change the line labels either to the letters "beta" and "alpha" or to the words "backscatter " and "extinction".*

Done!

**Response to Anonymous Referee #2**

**General questions:**

What do you think is the required power of a lidar system to perform Raman measurements at 1064 nm?

This cannot easily be answered. Laser power, size of the receiver telescope, receiver optics efficiency, and especially detector photon quantum efficiency are essential in this discussion. So, no easy answer! But we introduced a quite long new subsection 3.1 where we discuss all these aspects for two real-world lidars, our BERTHA lidar and the NASA GSFC lidar, they are very different concerning photon quantum efficiency (BERTHA, 0.08% at 1064 nm, NASA lidar, 38%). So, very different measurement scenarios regarding temporal and vertical resolution and accuracy in aerosol extinction profiling are described in section 3.1.

p. 3, l. 21: You should add the reference to Fig. 3 already here.

**Probably done now, we rephrased the text completely.**

Figure 3: Error bars for measurements at 355 nm and HSRL not visible. Error bars including the retrieval uncertainties (e.g. caused by the use of modeled temperature profiles) should be added.

All Raman lidar profiles have now error bars. We leave it out to put error bars to the HSRL solutions.

Error bars contain only the statistical signal noise errors, as usual...., so no uncertainty caused by the temperature profile is included... We provide the uncertainties (numbers) caused by temperature and pressure profile input in the figure caption of Figure 3. We checked the impact and found that the influence on extinction and lidar ratio profiling is almost negligible.

*p. 3, l. 24: Are the smaller values at 355 nm significant if you consider all measurement and retrieval uncertainties?*

Yes! Cannot be explained by signal noise or wrong atmospheric input parameter assumptions.

p. 3, l. 25: If the HSRL measurement is not discussed, they can be skipped in this paper.

HSRL results are not skipped. They are just in the background, will not confuse the reader, we believe.

p. 4, l. 12: What do you mean by 'most trustworthy height range'?

Is removed.

p. 4, l. 13: Change 'range from' to 'range of'

**Text is rephrased.**

p. 4, I 14: Change 'range fo' to 'range of'

**Text is rephrased.**

**Response to Dave Turner**

1) Line 26: the interference filter is 9 nm wide. This would seem to suggest that these observations are only useful during the night (with this interference filter). Is that correct? What are the prospects of making extinction measurements at 1064 nm using the Raman method in the daytime?

See the new subsection 3.1, and all the explanations. Subsection 3.1 was triggered by your comment! Thank you. To avoid a complex discussion, we just indirectly mention that one can use a comb filter as done by Arshinov, Appl. Opt. 2055, (Fabry Perot Interferometer, we cite this paper in sibsection 3.1), so daytime observations would be possible, too.

We discuss potential nighttime measurements with our lidar BERTHA used in this arcticle (photon counting, 0.08% quantum efficiency) and the NASA GSFC lidar of Dave Whiteman (APD, analog detection, 38% quantum efficiency) used in Veselovskii et al. (2015). Reasonable uncertainties in the retrieved extinction profile and temporal resolution of 1 hour and vertical resolution of around 750 m is only possible with the GSFC lidar.

However, and as already stated above: A simple answer is not possible. Therefore, we just compared to real-world operationa lidars. Laser power, size of the receiver telescope, receiver optics efficiency, and especially detector photon quantum efficieny is essential in this discussion. We have to wait for aerosol observations and experiments we will do in future with a Polly lidar, with clear focus on aerosols in the lower troposphere.

2) Signal-to-noise ratio will be an important limiter in this technique. If there is too little extinction over a layer (or the overall optical depth is too small), then it would be difficult to measure a good extinction profile. However, many aerosol layers have optical depths ranging from 0.1 to 0.25 (at 532 nm), but these layers are often closer to the lidar too. What sort of temporal averaging would be needed for these types of layers to get an accurate extinction profile?

See the explanations in Section 3.1, please.

**1064 nm rotational Raman lidar for particle extinction and lidar ratio profiling: Cirrus case study**

Moritz Haarig1, Ronny Engelmann1, Albert Ansmann1, Igor Veselovskii2, David N. Whiteman3, and Dietrich Althausen1

1Leibniz Institute for Tropospheric Research, Leipzig, Germany
 2Physics Instrumentation Center, Moscow, Russia
 3NASA, GSFC, Greenbelt, Maryland, USA

*Correspondence to:* M. Haarig (haarig@tropos.de)

**Abstract.** For the first time, vertical profiles of the 1064 nm particle extinction coefficient obtained from Raman lidar observations at 1058 nm (nitrogen and oxygen rotational Raman backscatter) are presented. We applied the new technique in the framework of test measurements and performed several cirrus observations of particle backscatter and extinction coefficients, and corresponding extinction-to-backscatter ratios at the wavelengths of 355, 532m and 1064 nm. The cirrus backscatter coef-

5 ficients were found to be equal for all three wavelengths keeping the retrieval uncertainties in mind. The multiple-scatteringcorrected cirrus extinction coefficients at 355 nm were on average about 20-30% lower than the ones for 532 and 1064 nm. The cirrus-mean extinction-to-backscatter ratio (lidar ratio) was  $31\pm5$  sr (355 nm),  $36\pm5$  sr (532 nm), and  $38\pm5$  sr (1064 nm) in this single study. We further discussed the requirements needed to obtain aerosol extinction profiles in the lower troposphere at 1064 nm with good accuracy (20 % relative uncertainty) and appropriate temporal and vertical resolution.

**10 1 Introduction**

Routine, height-resolved observations of the particle extinction coefficient in the atmosphere are only possible with lidar (Ansmann and Müller, 2005). The particle extinction coefficient is one of the key parameter in the description of the impact of clouds and aerosols on environmental, weather, and climate processes, and in the retrieval of microphysical properties of the detected cloud and aerosol layers. Two different lidar techniques are available for the measurement of aerosol and cloud extinction profiles, the Raman lidar method (Ansmann et al., 1990, 1992a; Ansmann and Müller, 2005) and the High Spectral

- 15 extinction profiles, the Raman lidar method (Ansmann et al., 1990, 1992a; Ansmann and Müller, 2005) and the High Spectral Resolution Lidar (HSRL) technique (Shipley et al., 1983; Grund and Eloranta, 1990; Hair et al., 2001; Eloranta, 2005). An HSRL is measuring strong Rayleigh backscatter signals so that likewise short signal averaging periods are sufficient to retrieve high-quality extinction profiles. The technique is thus well suited for airborne and spaceborne applications (Wandinger et al., 2002; Hair et al., 2008; Esselborn et al., 2008; Burton et al., 2012; Illingworth et al., 2015). The Raman lidar method is based
- 20 on comparably weak nitrogen and oxygen Raman backscatter lidar returns so that longer signal averaging times are required for an accurate extinction profiling. However, the lidar setup is relativley simple and robust and therefore the Raman lidar technique has proven to be of advantage for long-term ground-based measurements (Turner et al., 2001; Pappalardo et al.,

2014; Baars et al., 2016). This article deals with first atmospheric measurements with a novel rotational Raman channel at 1058 nm implemented in an operational Raman lidar to allow extinction profiling simultaneously at 355, 532, and 1064 nm.

The Raman lidar technique is used to measure particle backscatter and extinction coefficients, and the extinction-to-backscatter ratio (lidar ratio) since more the 25 years (Ansmann et al., 1990, 1992a, 1993; Ferrare et al., 1993, 1998). Single-wavelength

- Raman lidars transmitted laser pulses at 308 nm or 355 nm, and recorded height profiles of signals elastically backscat-5 tered by air molecules and particles (at 308 or 355 nm) and inelastically (Raman) backscattered by nitrogen molecules at 332 nm or 387 nm (vibrational-rotational spectrum), respectively. In the next step, dual-wavelength Raman lidar came into operation (Müller et al., 1998; Ansmann et al., 2000; Mattis et al., 2002; Wandinger et al., 2002; Murayama et al., 2004) with laser wavelengths at 355 and 532 nm and respective vibrational-rotational nitrogen Raman channels at 387 and 607 nm. The
- aerosol Raman lidar technique opened a solid way for comprehensive studies of anthropogenic pollution, desert dust, vol-10 canic dust, and biomass burning smoke at several wavelengths (e.g., Mattis et al., 2004; Müller et al., 2005; Tesche et al., 2009a; Alados-Arboledas et al., 2011; Groß et al., 2011, 2012; Nicolae et al., 2013; Kanitz et al., 2014; Veselovskii et al., 2016) and found broad application in the European Aerosol Research Lidar Network, EARLINET (e.g., Amiridis et al., 2005; Papayannis et al., 2008; Mona et al., 2014; Pappalardo et al., 2014). Particle extinction profiling is also a basic requirement for
- a successful retrieval of microphysical aerosol properties by means of inversion methods (Müller et al., 1998; Müller et al., 15 2000; Müller et al., 2013; Ansmann and Müller, 2005; Veselovskii et al., 2002, 2016).

However, it remained an open issue throughout the years to measure the aerosol extinction coefficient at 1064 nm. Stateof-the-art multiwavelength aerosol lidars use Nd:YAG lasers transmitting laser pulses at 355, 532 and 1064 nm. Techniques have been developed to determine height profiles of particle backscatter and extinction coefficients at 355 and 532 nm, but not at 1064 nm. Only the 1064 nm backscatter coefficient is delivered. The improved coverage of the wavelength spectrum by

20

backscatter and extinction pairs at all three Nd:YAG wavelengths would permit an improved, more robust characterization of aerosols in terms of optical and microphysical properties with lidar.

25

Recently, Veselovskii et al. (2015) presented high-quality aerosol measurements with an interference-filter-based 532 nm rotational Raman lidar. This successful feasibility study paved the way to design a 1058 nm interference filter concept for successful particle extinction profiling at 1064 nm. In this article, we report the first lidar observations of particle extinction at 1064 nm by using such 1058 nm interference filters. The implementation of a 1058 nm rotational Raman channel into an operational 355/532 nm Raman lidar is presented. A first three-wavelength Raman lidar measurement performed in a thick cirrus layer is shown and discussed. In addition, the potential of the new method to allow for aerosol extinction profiling in the lower troposphere with appropriate temporal and vertical resolution and accuracy is discussed.

**2 Instrumentation 30**

The methodological background of particle backscattering and extinction profiling with rotational and vibration-rotational Raman lidars was recently reviewed by Veselovskii et al. (2015) and is partly based on the studies of Whiteman (2003a, b). The theoretical framework, presented with focus on the Raman lidar applications at 355 and 532 nm wavelength, can be used for the retrieval of particle backscattering and extinction at 1064 nm wavelength as well.

We implemented the new 1058 nm rotational Raman channel in our containerized multi-wavelength polarization/Raman lidar BERTHA (Backscatter Extinction lidar-Ratio Temperature Humidity profiling Apparatus) (Althausen et al., 2000; Haarig et al.,

- 5 2015). BERTHA has been used over the past 20 years in 12 major field campaigns in Europe, Asia, Africa and the Caribbean (Wandinger et al., 2002; Ansmann et al., 2002; Franke et al., 2003; Tesche et al., 2009b; Tesche et al., 2011; Haarig et al., 2015). In 2012, BERTHA was re-designed to allow particle linear depolarization measurements at 355, 532, and 1064 nm, simultaneously. During our last field campaign, the Saharan Aerosol Long-Range Transport and Aerosol-Cloud-Interaction Experiment SALTRACE (Barbados, 4-week campaigns the summers of 2013 and 2014, and a 3-week-campaign in the winter
- 10 of 2014), BERTHA enabled us to derive vertical profiles of particle linear depolarization ratio and backscatter coefficients at three wavelength and, by means of measured vibration-rotational Raman signals, of particle extinction and extinction-tobackscatter ratio (lidar ratio) at two wavelengths (355 and 532 nm). BERTHA permits us in addition to measure 532 nm particle extinction profiles by means of the HSRL technique at 532 nm (Althausen et al., 2012). The complex lidar setup will be described in detail in the SALTRACE special issue of ACP (Haarig et al., 2016, in preparation).
- Two Nd: YAG lasers (Continuum, Powerlite) transmit linearly polarized laser pulses at 355 and 1064 nm (first laser) and at 532 nm (second laser). Two additional linear polarizers clean the polarization of the outgoing light. The repetition rate is 30 Hz and the pulse length 8–10 ns. The pulse energies can be as high as 1000 mJ (1064 nm), 800 mJ (532 nm) and 120 mJ (355 nm) in the ideal case of well-working optical elements in the transmission unit of the lidar. However, the pulse energies were only about 50% of these maximum values during the October 2015 measurements. The beams are expanded tenfold and
- 20 pointed into the atmosphere at an off-zenith angle of 5° to avoid the influence of specular reflection by ice crystals in cirrus layers on the backscattered signals. The receiver field of view (RFOV) is 0.8 mrad. A 53 cm Cassegrain telescope collects the backscattered light. The signals are detected with a range resolution of 7.5 m and a time resolution of 10 s. To avoid overloading of the photomultipliers (PMTs, photon counting mode) in the near range we restricted the maximum count rate to 20 MHz for the signal maximum in about 500 m height. This was achieved by using neutral density filters in front of the photomultipliers.
- 25 The reduction of the backscatter photon strength is small for the Raman channels. No neutral density filter was used in the 1058 nm Raman channel in the measurement presented in the next section. We further checked a potential dependence of the transmission characteristics of the 1058 nm Raman channel (beam steering receiver optics) on the state of linear polarization of incoming (cirrus or dust) backscatter photons and found no dependency. Systematic effects caused by the receiver optics on the retrieval of cirrus optical properties (in the next section) can be excluded.
- 30 A sketch of the receiver unit is given in Fig. 1. We replace the 1064 nm interference filter in the total 1064 nm backscatter signal channel by the 1058 nm interference filter (now used as the 1058 nm rotational Raman channel), and we removed the linear sheet polarizer of the 1064 nm cross-polarized signal channel and used this channel as the total 1064 nm backscatter signal channel. As mentioned, the design and realization of the 1058 nm interference filter used for 1064 nm backscatter and extinction profiling was motivated by the successful demonstration of high-quality aerosol measurements with an interference-filter-
- 35 based 532 nm rotational Raman lidar (Veselovskii et al., 2015). The same methodological approach, for the laser wavelength

of 532.07 nm, is applied here for the wavelength of 1064.14 nm. The interference filter is centered at about 1058 nm (Alluxa, Santa Rosa, CA, http://www.alluxa.com). The transmission band is from 1053-1062 nm with a transmission >90% in this wavelength range. For the laser wavelength of 1064.24 nm, the transmission is specified to be 0.005%. A broad band blocking by more than four orders magnitude from 350-1100 nm is realized. All filter-related transmission and blocking specifications are taken from the manufacturer's data sheets. provided by Alluxa

5 are taken from the manufacturer's data sheets, provided by Alluxa.

When using a laser wavelength of 1064.14 nm, 88% of the total intensity of the rotational Raman backscatter spectrum (anti-Stokes lines) can pass the 1053-1062 nm filter. The filter width of 9 nm restricts the detection of 1058 nm Raman signals to nighttime sky background conditions. The detected Raman backscatter intensity is only weakly temperature-dependent (4% increase of the measured rotational Raman signal for a temperature decrease from 300 K to 230 K). To guarantee no cross-talk

10

20

25

contributions in the rotational Raman channel even in the case of extremely high elastic backscattering in ice clouds, we used two Raman filters in front of the PMT.

With these two filters, we achieved an overall blocking of elastic backscatter light in the 1058 nm Raman channel by 2.5 times  $10^{-5}$ . According to Ansmann et al. (1992b), the Raman filters must be able to suppress elastic backscatter signals resulting from specular reflection by ice cystrals by up to 5–6 orders of magnitude, when pointing to the zenith. Operation of the lidar

15 at an off-zenith angle of 5° reduces the necessary suppression to about  $10^{-4}$  because specular reflection is not detectable with this tilted lidar.

The data analysis is performed following the EARLINET data analysis protocol (Pappalardo et al., 2004). In the computation of Rayleigh backscattering and extinction contributions to the measured lidar signals, we used GDAS (Global Data Assimilation System) height profiles of temperature and pressure of the National Weather Service's National Centers for Environmental Prediction (NCEP, NOAA's Air Resources Laboratory ARL, https://www.ready.noaa.gov/gdas1.php).

**3 Results**

First test measurements with the new Raman channel were performed on 5 October 2015. Fig. 2 presents an observation performed on 12 October with a well aligned lidar during a long-lasting cirrus event. Cirrus provides almost optimum conditions for checking the performance of the new lidar approach. The overlap between the laser beam and the receiver field of view (RFOV) is complete in the far range of the lidar and does not introduce any bias in the retrieval products. Furthermore, the

- suppression of elastic backscatter light in the used 1058 nm Raman channel (anti-Stokes rotational Raman signal passing a filter centered at 1058 nm) can be checked at extremely large ice-crystal backscatter conditions. In cirrus, the particle backscatter, extinction, and lidar ratio profiles for all three wavelength (355, 532, and 1064 nm) should be rather similar because of the expected almost wavelength-independent backscattering and extinction properties when the laser wavelengths are small
- 30 compared to the size of ice crystals. In Fig. 3, we averaged 216000 laser shots (transmitted within two hours). To reduce the uncertainty in the retrieval products caused by signal noise to a tolerable level of 20%, we smoothed the temporally averaged signal profiles in the retrieval of the cirrus particle backscatter coefficient with a vertical window length of 200 m. The extinction profiles in Fig. 3 were obtained from a least-squares linear regression analysis applied to the 387, 607, and 1058 nm

4

Raman signal profiles. The window lengths are 2000 m in the case of 355 nm and 532 nm, and 2500 m and 3200 m below and above 9 km height, respectively in the case of 1064 nm and the HSRL extinction profile. The lidar ratios are calculated for the same coarse height resolution as the extinction values. Multiple scattering effects are not corrected in Fig. 3. The impact will be discussed below. The calibration of the backscatter coefficient profiles were performed within the height range just

above the cirrus layer (12-15 km height, 355 and 532 nm) or below the ice cloud (between 4 and 5.5 km height, 1064 nm). 5 The particle coefficient was set to  $10^{-2}$  Mm-1sr-1 at all three wavelength. This means that Rayleigh scattering determines the total calibration backscatter coefficient in the calibration height range for 355 and 532 nm.

As can be seen in Fig. 2, the ice cloud layer showed a stable base height at 6.5 km. The top height was detected at 10.5 km, probably coinciding with the tropopause. The backscatter intensity increased by two orders of magnitude at cloud base according to the 1064 nm elastic backscatter signal (right panel in Fig. 2). As mentioned, to avoid strong specular reflection effects 10

15

the laser beams were pointing to an off-zenith angle of  $5^{\circ}$ . The Raman signal profiles do not show any interference by strong elastic backscatter by ice crystals. The 4 km deep, comparably homogeneous cirrus layers provided favorable conditions for a proper determination of the

cirrus optical properties. The two-hour mean height profiles of particle backscatter and extinction coefficients, and lidar ratio at 355, 532, and 1064 nm are shown in Fig. 3. To present the full potential of the BERTHA lidar for aerosol and cloud research, HSRL solutions (for 532 nm) are shown as well.

Keeping the uncertainty bars into consideration, similar profiles of the particle backscatter coefficients are obtained at all three wavelengths with all applied methods (Raman lidar and HSRL methods). All profiles are based on the analysis of the profiles of the ratio of the elastic backscatter signal to the molecule signal.

In the case of the extinction profiles, a good agreement between the 532 and 1064 nm extinction profiles (Raman lidar 20 solutions) is visible. The 1064 nm extinction profile is however comparably more noisy. The 355 nm extinction coefficients are considerably smaller (by 20-40 Mm-1, about 20–25%) than the 532 nm extinction values. In the case of the 532 nm HSRL solutions not all necessary corrections were applied (e.g., proper consideration of all cross-talk effects and near-range iodineabsorption saturation effects). This explains the deviations from the other profiles. We will use this case in a follow-up paper which will focus explicitly on the performance of the HSRL branch of BERTHA. 25

Multiple scattering (MS) effects are not corrected in Fig. 3. According to Seifert et al. (2007), the MS factor, defined as the apparent (observed) extinction coefficient divided by the respective single-scattering (SS) extinction coefficient, is 0.6 at cirrus base and larger than 0.9 at cloud top for BERTHA with an RFOV of 0.8 mrad. The computations were performed for 532 nm. MS effects are caused by strong forward scattered laser radiation which remains in the RFOV so that the effective attenuation

- of laser radiation by ice crystal scattering is significantly reduced and leads to an apparent (effective) extinction coefficient and 30 lidar ratio, about a factor of 1.1-1.8 lower than the respective single-scattering values. For Fig. 3, we can conclude, that the MS-corrected extinction values are close to  $200 \text{ Mm}^{-1}$  (355 nm) and 250 Mm-1 (532 nm) around 7–8 km height (cirrus base), and 85 Mm-1 (355 nm) and 110 Mm-1 (532 nm, 1064 nm) around 9–10 km height (cirrus top). Here we assume, that MS effects are wavelength-independent. The cirrus SS-related optical depth is close to 0.46 at 355 nm, 0.59 at 532 nm, and around
- 0.6 at 1064 nm when considering the extinction values from the Raman lidar observations between 6 and 10 km height. 35

Reasons for the deviation of the 355 nm extinction profile could be an increase in the MS effect with decreasing wavelength. Another reason could be related to the cirrus crystal size distribution which caused stronger extinction at 532 and 1064 nm than at the shorter wavelength of 355 nm. However, accompanying Doppler lidar observations of crystal terminal velocity (about 0.7 m/s at cloud base) indicate large crystals (with diameters typically exceeding 300-400  $\mu$ m) so that a size-related wavelength dependence of backscattering and extinction is not very likely.

The lidar ratio observations at three wavelength in the right panel in Fig. 3 are the highlight of the study. The lidar ratio profiles show similar vertical structures for all three wavelengths. The slight differences between the effective lidar ratios (not corrected for the MS effect) may point to the different impact of MS for the different wavelengths. The cirrus lidar ratios for 532 and 1064 nm are almost equal with values around 22.5–25 sr at cirrus base to about 35–40 sr at cirrus top. The 355 nm values are 3–5 sr lower than the 532 nm lidar ratios. Taking the MS effect into account with an average MS factor of about 0.8

10 values are 3–5 sr lower than the 532 nm lidar ratios. Taking the MS effect into account with an average MS factor of about 0.8 for the 4–km deep cirrus layer and we end up with MS corrected cirrus mean lidar ratios around 31 sr for 355 nm, 36 sr for 532 nm, and 38 for 1064 nm for this single cirrus event. The variability around these mean values is of the order of 5 sr.

There are numerous reports on cirrus lidar ratios in the literature for comparison. However, clear statements on MS correction and laser pointing (zenith pointing or off-zenith pointing) are often missing so that comparisons are difficult. Seifert et al.

- 15 (2007) reported BERTHA lidar observation of the cirrus clouds over the tropical Indian Ocean (Indian Ocean Experiment, 1999–2000). During the Northeast monsoon season (polluted winter season) the MS-corrected mean cirrus lidar ratio was 33±9 sr at 532 nm for off-zenith pointing conditions and for cirrus layers between 9 and 18 km height. Chen et al. (2002) reported MS-corrected 532 nm cirrus lidar ratios of 35±15 sr for the 12-15 km (comparably warm ice cloud) range over Taiwan for the years 1999 ans 2000 obtained with an obviously zenith-pointing lidar. Giannakaki et al. (2007) used an off-
- 20 zenith pointing 355 nm backscatter lidar and found MS-corrected lidar ratios of  $30\pm17$  sr in cirrus clouds over northern Greece in the Mediterranean for the 2002-2006 time period. Cirrus were observed between 9 and 13 km height with mid cloud temperatures from -40 to  $-65^{\circ}$ C. Josset et al. (2012) and Garnier et al. (2015) analyzed spaceborne CALIOP (Cloud Aerosol Lidar with Orthogonal Polarization) lidar observations, which were partly performed at zenith-pointing and off-zenith-pointind conditions. The authors concluded that the MS-corrected cirrus lidar ratio around the globe are typically 30-35 sr  $\pm$ 5-8 sr at
- 25 532 nm. Garnier et al. (2015) found that the cirrus mean lidar ratios are on average around 35 sr for comparably warm ice clouds with temperatures higher than about  $-45^{\circ}$ C. The cirrus layer, discussed in Fig. 3, showed temperatures from  $-25^{\circ}$ C at 6.5 km height over  $-36^{\circ}$ C at 8 km height to about  $-50-55^{\circ}$ C at cirrus top. Thus our observation fits well into the published lidar ratio climatologies. Furthermore, the simultaneous 532 and 1064 nm Raman lidar observations, which suggest wavelengthindependent cirrus backscattering, extinction, and lidar ratio, corroborate that all the assumption which have to be made in the
- 30 CALIOP cirrus data analysis regarding wavelength-dependence of cirrus optical properties in the 532-1064 nm spectral range are justified (Vaughan et al., 2010).

**3.1 Outlook: aerosol particle extinction profiling at 1064 nm**

5

As mentioned above, the new method in its simplest configuration, without daylight suppression between the rotational Raman lines as suggested by Arshinov et al. (2005), is only applicable at nighttime hours because of the broad 9 nm interference filter

width. In the following discussion, we illuminate the potential of two operational lidars regarding 1064 nm extinction profiling of aerosol layers in the lower troposphere (around 1.5 km height). These two lidars are BERTHA (with the key features: about 500 mJ pulse energy at 1064 nm, 30 Hz repetition rate, 53 cm telescope, photon counting with single-photon quantum efficiency of 0.08%) and the NASA GSFC multiwavelength Raman lidar (150 mJ pulse energy at 532 nm, 50 Hz repetition rate,

40 cm telescope, analog detection with avalanche photodiode, single-photon quantum efficiency of close to 40%) described in 5 Veselovskii et al. (2013). We discuss scenarios with 1064 nm aerosol extinction values around 150  $Mm^{-1}$  (lofted mineral dust layer) and 50  $Mm^{-1}$  (more typical for smoke and pollution layers).

From our first cirrus observation with the new 1064 nm rotational Raman channel in the BERTHA system discussed above we can draw the following conclusions regarding aerosol extinction profiling at 1.5 km height. In Fig. 3, at 8–8.5 km height the

- measured 1064 nm extinction coefficient is 160  $Mm^{-1}$  (relative uncertainty 20%). At 1.5 km height, the signal strength would 10 be a factor of about 30 higher than at 8 km height due to the reduced distance from the lidar. By using the same temporal and vertical resolution as in Fig. 3, the relative uncertainty would thus decrease by roughly a factor of 5.5 ( $\sqrt{30}$ ). To obtain aerosol extinction coefficients with an acceptable uncertainty of 20% (as in the cirrus case in Fig. 3), we can therefore reduce the smoothing length  $\Delta R$  from 2500 m to about 800 m. The uncertainty decreases with  $1/(\Delta R)^{3/2}$  (Browell et al., 1979) when
- the solution follows from a difference quotient with the step with of  $\Delta R$  as is the case in the extinction-coefficient retrieval. 15 However, if the extinction coefficient is 50  $Mm^{-1}$  (typical for smoke and haze), we need a regression window length of 1600 m to keep the relative uncertainty close to 20%. This means, even by using a powerful 1064 nm lidar such as BERTHA, the photon counting unit (with a PMT quantum efficiency of 0.08%) is not appropriate for aerosol extinction profiling in the lower troposphere. Large signal averaging periods in combination with very large vertical regression window lengths are required to keep the uncertainties at a tolerable level of 20%.

20

The NASA GSFC lidar makes use of analog detection at 1064 nm, with a quantum efficiency a factor of almost 500 higher than the quantum efficiency (photon counting mode) of BERTHA. Veselovskii et al. (2015) presents a 532 nm rotational Raman lidar measurement in terms of aerosol extinction, backscatter, and lidar ratio profiles in the lower troposphere. 30 minutes of signal averaging was sufficient to obtain the particle extinction coefficients of about 150  $Mm^{-1}$  at 1.5 km height with a vertical

- resolution of 100 m and a relative uncertainty of 5–10%. If we take the  $\lambda^{-4}$  wavelength dependence of molecular scattering 25 in the atmosphere into account, the rotational Raman signal strength in the 1064 nm wavelength range will be reduced by a factor of 16. For simplicity, we ignore here changing overall transmitter and receiver transmission features when going from 532 to 1064 nm. To keep the uncertainties below 10%, we need to increase the signal averaging period by a factor of 2 (to one hour signal averaging) and the vertical resolution must be decreased from 100 to 500 m. If we further assume that the 1064 nm aerosol extinction coefficient is of the order of 50  $Mm^{-1}$ , we still can have a reasonable resolution of about 750 m if
- 30

we acceptable a higher uncertainty of about 20%.

In the next step, we will implement a 1064 nm rotational Raman channel in a Polly system (Engelmann et al., 2016) and will make extensive aerosol observations with different detection methods. Both, analog and photon-counting detection and combinations of both need to be checked and tested. Note that not only extinction profiling is of value. Lidar ratio profiling

at 1064 nm and thus backscatter coefficient profiling is of importance, too. There are practically no experimental data on the 35

lidar ratio of the different dust types for 1064 nm. Almost all assumed values in the respective 1064 nm backscatter lidar retrievals rely on model computations. For an adequate backscatter profiling, the detection unit of the lidar receiver must be able to resolve six orders of magnitude of signal strength (from 1064 nm dust backscattering to almost pure 1064 nm Rayleigh backscattering) to allow proper calibration of the backscatter coefficient profiles. This is not possible by the use of analog detection.

**4 Conclusions**

5

10

We implemented a rotational Raman channel around 1058 nm in an operational mutiwavelength polarization/Raman lidar in order to obtain measured particle extinction coefficient profiles at 1064 nm. First measurements were performed in cirrus layers in October 2015. As expected, wavelength independent cirrus backscattering was observed at all three wavelength and also almost wavelength-independent extinction and lidar ratio at 532 and 1064 nm.

We discussed to what extend a proper 1064 nm extinction profiling and with appropriate temporal and range resolution is possible in the case of the TROPOS lidar BERTHA and the NASA GSFC Raman lidar. We concluded that efficient photon detection is required of obtain aerosol extinction and lidar ratio profiles with 10-20% relative uncertainty, one hour temporal resolution, and vertical resolution of 750 m.

- We conclude that in the next step, lidar observations of aerosol layers with different photon detection techniques (photon counting, analog detection, combination of both) is required with the goal to find an optimized detection system for a 1064 nm rotational Raman lidar for aerosol extinction and lidar ratio profiling. For proper extinction profiling at 1064 nm, analog detection seems to be of advantage. However, an optimized receiver optics and signal detection concept for a 1058 nm Raman channel needs to be elaborated based on an extended study with an operational continuously running lidar such as
- 20 Polly. Furthermore, Raman lidar observations at three wavelength should focus on aerosol layers (boundary layer, lower free troposphere) at very different conditions in terms of contribution by urban haze, biomass burning smoke, marine particles, and desert dust. Emphasis should also be put on the comparison with accompanying Aerosol Robotic Network (AERONET) photometer observations from 340-1640 nm as recently demonstrated by Veselovskii et al. (2016) by using a state-of-the-art dual-wavelength Raman lidar.
- 25 Acknowledgements. We thank Johannes Bühl for providing Doppler lidar observations of vertical velocity and estimated ice crystal sizes. Modeling of the rotational Raman filter parameters was supported by the Russian Science Foundation (project No. 16-17-10241).

**References**

- Alados-Arboledas, L., Müller, D., Guerrero-Rascado, J. L., Navas-Guzmán, F., Pérez-Ramírez, D., and Olmo, F. J.: Optical and microphysical properties of fresh biomass burning aerosol retrieved by Raman lidar, and star-and sun-photometry, Geophys. Res. Lett., 38, L01807, doi:10.1029/2010GL045999, 2011.
- 5 Althausen, D., Müller, D., Ansmann, A., Wandinger, U., Hube, H., Clauder, E., and Zörner, S.: Scanning six-wavelength eleven-channel aerosol lidar, J. Atmos. Oceanic Technol., 17, 1469-1482, doi:10.1175/1520-0426(2000)017<1469:SWCAL>2.0.CO;2, 2000.

Althausen, D., Oelsner, P., Rohmer, A., and Baars, H.: Comparison of High Spectral Resolution Lidar with Raman lidar, in *Reviewed and revised papers of the 26th International Laser Radar Conference*, A. Papayannis, D. Balis, and V. Amiridis (Eds.), Vol. 1, pages 43-46, June 2012, Porto Heli, Greece, National Technical University of Athens, 2012.

10 Amiridis, V., Balis, D. S., Kazadzis, S., Bais, A., Giannakaki, E., Papayannis, A., and Zerefos, E.: Four-year aerosol observations with a Raman lidar at Thessaloniki, Greece, in the framework of European Aerosol Research Lidar Network (EARLINET), J. Geophys. Res., 110, D21203, doi:10.1029/2005JD006190, 2005.

Ansmann, A., Riebesell, M., and Weitkamp, C.: Measurements of atmospheric aerosol extinction profiles with a Raman lidar, Opt. Letts., 15,

15 746-748, doi:10.1364/OL.15.000746, 1990.

- Ansmann, A., Wandinger, U., Riebesell, M., Weitkamp, C., and Michaelis, W.: Independent measurement of extinction and backscatter profiles in cirrus clouds by using a combined Raman elastic-backscatter lidar, Appl. Opt., 31, 7113-7131, 1992a.
- Ansmann, A., Riebesell, M., Wandinger, U., Weitkamp, C., Voss, E., Lahmann, W., and Michaelis, W.: Combined Raman elasticbackscatter LIDAR for vertical profiling of moisture, aerosol extinction, backscatter, and LIDAR ratio, Appl. Phys. B, 55, 18–22, doi:10.1007/BF00348608, 1992b
- Ansmann, A., Wandinger, U., and Weitkamp, C.: One-year observations of Mount-Pinatubo aerosol with an advanced Raman lidar over Germany at 53.5°N, Geophys. Res. Lett., 20, 711–714, doi:10.1029/93GL00266, 1993
- Ansmann, A., Althausen, D., Wandinger, U., Franke, K., Müller, D., Wagner, F., Heintzenberg, J.: Vertical profiling of the Indian aerosol plume with six-wavelength lidar during INDOEX: A first case study, Geophys. Res. Lett., 27, 963–966, doi:10.1029/1999GL010902,

25 2000.

20

- Ansmann, A., Wagner, F., Müller, D., Althausen, D., Herber, A., von Hoyningen-Huene, W., and Wandinger, U.: European pollution outbreaks during ACE 2: Optical particle properties inferred from multiwavelength lidar and star-Sun photometry, J. Geophys. Res., 107, doi:10.1029/2001JD001109, 2002.
- Ansmann, A., and Müller, D., Lidar and atmospheric aerosol particles, in LIDAR Range–resolved optical remote sensing of the atmosphere,

30 C. Weitkamp, ed., Springer, New York, 105–141, 2005.

- Arshinov, Y., Bobrovnikov, S., Serikov, I., Ansmann, A., Wandinger, U., Althausen, D., Mattis, I., and Müller, D.: Daytime operation of a pure rotational Raman lidar by use of a Fabry-Perot interferometer, Appl. Opt., 44, 3593-3603, doi: 10.1364/AO.44.003593, 2005.
- Baars, H., Kanitz, T., Engelmann, R., Althausen, D., Heese, B., Komppula, M., Preißler, J., Tesche, M., Ansmann, A., Wandinger, U., Lim, J.-H., Ahn, J. Y., Stachlewska, I. S., Amiridis, V., Marinou, E., Seifert, P., Hofer, J., Skupin, A., Schneider, F., Bohlmann, S., Foth, A.,
- Bley, S., Pfüller, A., Giannakaki, E., Lihavainen, H., Viisanen, Y., Hooda, R. K., Pereira, S. N., Bortoli, D., Wagner, F., Mattis, I., Janicka,
   L., Markowicz, K. M., Achtert, P., Artaxo, P., Pauliquevis, T., Souza, R. A. F., Sharma, V. P., van Zyl, P. G., Beukes, J. P., Sun, J., Rohwer,

E. G., Deng, R., Mamouri, R.-E., and Zamorano, F.: An overview of the first decade of PollyNET: an emerging network of automated Raman-polarization lidars for continuous aerosol profiling, Atmos. Chem. Phys., 16, 5111-5137, doi:10.5194/acp-16-5111-2016, 2016.

- Browell, E., Wilkerson, T., and Mcilrath, T.: Water vapor differential absorption lidar development and evaluation, Appl. Opt. 18, 3474-3483, doi: 10.1364/AO.18.003474, 1979.
- 5 Burton, S. P., Ferrare, R. A., Hostetler, C. A., Hair, J. W., Rogers, R. R., Obland, M. D., Butler, C. F., Cook, A. L., Harper, D. B., and Froyd, K. D.: Aerosol classification using airborne High Spectral Resolution Lidar measurements – methodology and examples, Atmos. Meas. Tech., 5, 73-98, doi:10.5194/amt-5-73-2012, 2012.
  - Chen, W.-N., Chiang, C.-W., and Nee, J.-B.: Lidar ratio and depolarization ratio for cirrus clouds, Appl. Opt.,41, 6470-6476, doi: 10.1364/AO.41.006470, 2002.
- 10 Eloranta, E. W.: High spectral resolution lidar, in: Lidar: Range-resolved Optical Remote Sensing of the Atmosphere, edited by: Weitkamp, K., Springer, New York, 143–163, 2005.
  - Engelmann, R., Kanitz, T., Baars, H., Heese, B., Althausen, D., Skupin, A., Wandinger, U., Komppula, M., Stachlewska, I. S., Amiridis, V., Marinou, E., Mattis, I., Linné, H., and Ansmann, A.: The automated multiwavelength Raman polarization and water-vapor lidar PollyXT: the neXT generation, Atmos. Meas. Tech., 9, 1767-1784, doi:10.5194/amt-9-1767-2016, 2016.
- 15 Esselborn, M., Wirth, M., Fix, A., Tesche, M., and Ehret, G.: Airborne high spectral resolution lidar for measuring aerosol extinction and backscatter coefficients, Appl. Opt. 47, 346-358, doi:10.1364/AO.47.000346, 2008.
  - Ferrare, R. A., Melfi, S. H., Whiteman, D. N., and Evans, K. D.: Raman lidar measurements of Pinatubo aerosols over southeastern Kansas during November-December 1991, Geophys. Res. Lett., 19, 1599–1602, doi:10.1029/92GL01473, 1992
- Ferrare, R. A., Melfi, S. H., Whiteman, D. N., Evans, K. D., and Leifer, R.: Raman lidar measurements of aerosol extinction and backscattering: 1. Methods and comparisons, J. Geophys. Res., 103(D16), 19663–19672, doi:10.1029/98JD01646, 1998.
- Franke, K., Ansmann, A., Müller, D., Althausen, D., Venkataraman, C., Reddy, M. S., Wagner, F., and Scheele, R.: Optical properties of the Indo-Asian haze layer over the tropical Indian Ocean, J. Geophys. Res., 108, 4059, doi:10.1029/2002JD002473, D2, 2003.
  - Garnier, A., Pelon, J., Vaughan, M. A., Winker, D. M., Trepte, C. R., and Dubuisson, P.: Lidar multiple scattering factors inferred from CALIPSO lidar and IIR retrievals of semi-transparent cirrus cloud optical depths over oceans, Atmos. Meas. Tech., 8, 2759-2774, doi:10.5194/amt-8-2759-2015.2015
- doi:10.5194/amt-8-2759-2015, 2015.
  - Giannakaki, E., Balis, D. S., Amiridis, V., and Kazadzis, S.: Optical and geometrical characteristics of cirrus clouds over a Southern European lidar station, Atmos. Chem. Phys., 7, 5519-5530, doi:10.5194/acp-7-5519-2007, 2007.
  - Groß, S., Tesche, M., Freudenthaler, V., Toledano, C., Wiegner, M., Ansmann, A., Althausen, D., and Seefeldner, M.: Characterization of Saharan dust, marine aerosols and mixtures of biomass-burning aerosols and dust by means of multi-wavelength depolarization and Raman
- 30 lidar measurements during SAMUM 2, Tellus B, 63, 706-724, doi:10.1111/j.1600-0889.2011.00556.x, 2011.
  - Groß, S., Freudenthaler, V., Wiegner, M., Gasteiger, J., Geiß, A., and Schnell, F.: Dual-wavelength linear depolarization ratio of volcanic aerosols: Lidar measurements of the Eyjafjallajökull plume over Maisach, Germany, Atmos. Environ., 48, 85–96, 2012.
    - C. J. Grund, and Eloranta, E. W.: The 27–28 October 1986 FIRE IFO Cirrus Case Study: Cloud Optical Properties Determined by High Spectral Resolution Lidar. Mon. Wea. Rev. 118, 2344–2355, doi: 10.1175/1520-0493(1990)118<2344:TOFICC>2.0.CO;2, 1990.
- 35 Haarig, M., Althausen, D., Ansmann, A., Klepel, A., Baars, H., Engelmann, R., Groß, S., and Freudenthaler, V.: Measurement of the linear depolarization ratio of aged dust at three wavelengths (355, 532 and 1064 nm) simultaneously over Barbados, in: Proceedings of the 27th International Laser Radar Conference, New York City, 5–10 July 2015, S8b.04, 2015.

- Hair, J., Caldwell, L., Krueger, D., and She, C., High-Spectral-Resolution Lidar with Iodine-Vapor Filters: Measurement of Atmospheric-State and Aerosol Profiles, Appl. Opt. 40, 5280–5294, doi:10.1364/AO.40.005280, 2001.
- Hair, J. W., Hostetler, C. A., Cook, A. L., Harper, D. B., Ferrare, R. A., Mack, T. L., Welch, W., Izquierdo, L. R., and Hovis, F. E.: Airborne high-spectral-resolution lidar for profiling aerosol optical profiles, Appl. Opt., 47, 6734–6752, doi:10.1364/AO.47.006734, 2008.
- 5 Illingworth, A. J., Barker, H. W., Beljaars, A., Ceccaldi, M., Chepfer, H., Clerbaux, N., Cole, J., Delanoe, J., Domenech, C., Donovan, D. P., Fukuda, S., Hirakata, M., Hogan, R. J., Hünerbein, H., Kollias, P., Kubota, T., Nakajima, T., Nakajima, T. Y., Nishizawa, T., Ohno, Y., Okamoto, H., Oki, R., Sato, K., Satoh, M., Shephard, M., Velázquez-Blázquez, A., Wandinger, U., Wehr, T., and Zadelhoff, G.-J.: THE EARTHCARE SATELLITE: the next step forward in global measurements of clouds, aerosols, precipitation and radiation, B. Am. Meteorol. Soc., 96, 1311–1332, doi:10.1175/BAMS-D-12-00227.1, 2015.
- 10 Josset, D., Pelon, J., Garnier, A., Hu, Y., Vaughan, M., Zhai, P.-W., Kuehn, R., and Lucker, P.: Cirrus optical depth and lidar ratio retrieval from combined CALIPSO-CloudSat observations using ocean surface echo, J. Geophys. Res., 117, D05207, doi:10.1029/2011JD016959, 2012.
  - Kanitz, T., Engelmann, R., Heinold, B., Baars, H., Skupin, A., and Ansmann, A.: Tracking the Saharan Air Layer with shipborne lidar across the tropical Atlantic, Geophys. Res. Lett., 41, 1044–1050, doi:10.1002/2013GL058780, 2014.
- 15 Mattis, I., Ansmann, A., Müller, D., Wandinger, U., and Althausen, D.: Dual-wavelength Raman lidar observations of the extinction-tobackscatter ratio of Saharan dust, Geophys. Res. Lett., 29, 1306, doi:10.1029/2002GL014721, 2002.
  - Mattis, I., Ansmann, A., Müller, D., Wandinger, U., and Althausen, D.: Multiyear aerosol observations with dual-wavelength Raman lidar in the framework of EARLINET, J. Geophys. Res., 109, D13203, doi:10.1029/2004JD004600, 2004.
  - Mona, L., Papagiannopoulos, N., Basart, S., Baldasano, J., Binietoglou, I., Cornacchia, C., and Pappalardo, G.: EARLINET dust observa-
- 20 tions vs. BSC-DREAM8b modeled profiles: 12-year-long systematic comparison at Potenza, Italy, Atmos. Chem. Phys., 14, 8781-8793, doi:10.5194/acp-14-8781-2014, 2014.
  - Müller, D., Wandinger, U., Althausen, D., Mattis, I., and Ansmann, A.: Retrieval of physical particle properties from lidar observations of extinction and backscatter at multiple wavelengths, Appl. Opt., 37, 2260-2263, doi: 10.1364/AO.37.002260, 1998.

Müller, D., Wandinger, U., and Ansmann, A.: Microphysical particle parameters from extinction and backscatter lidar data by inversion with regularization: theory, Appl. Opt., 38, 2346-2357, doi:10.1364/AO.38.002346, 1999.

25

- Müller, D., Wagner, F., Althausen, D., Wandinger, U., and Ansmann, A.: Physical properties of the Indian aerosol plume derived from six-wavelength lidar observations on 25 March 1999 of the Indian Ocean Experiment, Geophys. Res. Lett., 27, 1403–1406, doi:10.1029/1999GL011217, 2000.
  - Müller, D., Mattis, I., Wandinger, U., Ansmann, A., Althausen, D., and Stohl, A.: Raman lidar observations of aged Siberian and Canadian
- 30 forest fire smoke in the free troposphere over Germany in 2003: Microphysical particle characterization, J. Geophys. Res., 110, D17201, doi:10.1029/2004JD005756, 2005.
  - Müller, D., Ansmann, A., Mattis, I., Tesche, M., Wandinger, U., Althausen, D., and Pisani, G.: Aerosol-type-dependent lidar ratios observed with Raman lidar, J. Geophys. Res., 112, D16202, doi:10.1029/2006JD008292, 2007.
  - Müller, D., Lee, K.-H., Gasteiger, J., Tesche, M., Weinzierl, B., Kandler, K., Müller, T., Toledano, C., Otto, S., Althausen, D., and Ansmann,
- 35 A.: Comparison of optical and microphysical properties of pure Saharan mineral dust observed with AERONET Sun photometer, Raman lidar, and in situ instruments during SAMUM 2006, J. Geophys. Res., 117, D07211, doi:10.1029/2011JD016825.

- Müller, D, Veselovskii, I, Kolgotin, A, Tesche, M, Ansmann, A, and Dubovik, O.: Vertical profiles of pure dust and mixed smoke-dust plumes inferred from inversion of multiwavelength Raman/polarization lidar data and comparison to AERONET retrievals and in situ observations, Appl. Opt., 52, 3178-3202, doi:10.1364/AO.52.003178, 2013.
- Murayama, T., Müller, D., Wada, K., Shimizu, A., Sekiguchi, M. and Tsukamoto, T.: Characterization of Asian dust and Siberian smoke with 5 multiwavelength Raman lidar over Tokyo, Japan in spring 2003, Geophys. Res. Lett., 31, doi: 10.1029/2004GL021105, 2004.
- Nicolae, D., Nemuc, A., Müller, D., Talianu, C., Vasilescu, J., Belegante, L., and Kolgotin, A.: Characterization of fresh and aged biomass burning events using multiwavelength Raman lidar and mass spectrometry, J. Geophys. Res. Atmos., 118, 2956-2965, doi:10.1002/jgrd.50324, 2013.

Papayannis, A., Amiridis, V., Mona, L., Tsaknakis, G., Balis, D., Bösenberg, J., Chaikovski, A., De Tomasi, F., Grigorov, I., Mattis, I.,

- 10 Mitev, V., Müller, D., Nickovic, S., Pérez, C., Pietruczuk, A., Pisani, G., Ravetta, F., Rizi, V., Sicard, M., Trickl, T., Wiegner, M., Gerding, M., Mamouri, R. E., D'Amico, G., and Pappalardo, G.: Systematic lidar observations of Saharan dust over Europe in the frame of EARLINET (2000-2002), J. Geophys. Res., 113, D10204, doi:10.1029/2007JD009028, 2008.
  - Pappalardo, G., Amodeo, A., Pandolfi, M., Wandinger, U., Ansmann, A., Bösenberg, J., Matthias, V., Amiridis, V., De Tomasi, F., Frioud, M., Iarlori, M., Komguem, L., Papayannis, A., Rocadenbosch, F., and Wang, X.: Aerosol lidar intercomparison in the framework
- of the EARLINET project. 3. Raman lidar algorithm for aerosol extinction, backscatter, and lidar ratio, Appl. Opt., 43, 5370-5385, 15 doi:10.1364/AO.43.005370, 2004.
  - Pappalardo, G., Amodeo, A., Apituley, A., Comeron, A., Freudenthaler, V., Linné, H., Ansmann, A., Bösenberg, J., D'Amico, G., Mattis, I., Mona, L., Wandinger, U., Amiridis, V., Alados-Arboledas, L., Nicolae, D., and Wiegner, M.: EARLINET: towards an advanced sustainable European aerosol lidar network, Atmos. Meas. Tech., 7, 2389–2409, doi:10.5194/amt-7-2389-2014, 2014.
- 20 Seifert, P., Ansmann, A., Müller, D., Wandinger, U., Althausen, D., Heymsfield, A. J., Massie, S. T., and Schmitt, C.: Cirrus optical properties observed with lidar, radiosonde, and satellite over the tropical Indian Ocean during the aerosol-polluted northeast and clean maritime southwest monsoon, J. Geophys. Res., 112, D17205, doi:10.1029/2006JD008352, 2007.
  - Shipley, S. T., Tracy, D. H., Eloranta, E. W., Trauger, J. T., Sroga, J. T., Roesler, F. L., and Weinman, J. A.: High Spectral Resolution Lidar to measure optical-scattering properties of atmospheric aerosols, 1. Theory and instrumentation, Appl. Opt., 22, 3716–3724, doi: 10.1364/AO.22.003716, 1983
- 25
  - Tesche, M., Ansmann, A., Müller, D., Althausen, D., Engelmann, R., Freudenthaler, V., and Groß, S.: Vertically resolved separation of dust and smoke over Cape Verde using multiwavelength Raman and polarization lidars during Saharan Mineral Dust Experiment 2008, J. Geophys. Res., 114, D13202, doi:10.1029/2009JD011862, 2009a.

Tesche, M., Ansmann, A., Müller, D., Althausen, D., Mattis, I., Heese, B., Freudenthaler, V., Wiegner, M., Eseelborn, M., Pisani, G., and

- 30 Knippertz, P.: Vertical profiling of Saharan dust with Raman lidars and airborne HSRL in southern Morocco during SAMUM, Tellus B, 61, 144–164, doi:10.1111/j.1600-0889.2008.00390.x, 2009b.
  - Tesche, M., Groß, S., Ansmann, A., Müller, D., Althausen, D., Freudenthaler, V., and Esselborn, M.: Profiling of Saharan dust and biomass-burning smoke with multiwavelength polarization Raman lidar at Cape Verde, Tellus B, 63, 649-676, doi:10.1111/j.1600-0889.2011.00548.x, 2011.
- 35 Turner, D. D., Ferrare, R. A., and Brasseur, L. A.: Average aerosol extinction and water vapor profiles over the Southern Great Plains, Geophys. Res. Lett., 28, 4441-4444, doi:10.1029/2001GL013691, 2001

- Vaughan, M. A., Liu, Z., McGill, M. J., Hu, Y., and Obland, M. D.: On the spectral dependence of backscatter from cirrus clouds: Assessing CALIOP's 1064 nm calibration assumptions using cloud physics lidar measurements, J. Geophys. Res., 115, D14206, doi:10.1029/2009JD013086, 2010.
- Veselovskii, I., Kolgotin, A., Griaznov, V., Müller, D., Wandinger, U., and Whiteman, D.,: Inversion with regularization for the retrieval of tropospheric aerosol parameters from multiwavelength lidar sounding, Appl. Opt., 41, 3685-3699, doi:10.1364/AO.41.003685, 2002.
- Veselovskii, I., Whiteman, D. N., Korenskiy, M., Kolgotin, A., Dubovik, O., Perez-Ramirez, D., and Suvorina, A.: Retrieval of spatiotemporal distributions of particle parameters from multiwavelength lidar measurements using the linear estimation technique and comparison with AERONET, Atmos. Meas. Tech., 6, 2671-2682, doi:10.5194/amt-6-2671-2013, 2013.

Veselovskii, I., Whiteman, D. N., Korenskiy, M., Suvorina, A., and Pérez-Ramírez, D.: Use of rotational Raman measurements in multiwave-

- 10 length aerosol lidar for evaluation of particle backscattering and extinction, Atmos. Meas. Tech., 8, 4111-4122, doi:10.5194/amt-8-4111-2015, 2015.
  - Veselovskii, I., Goloub, P., Podvin, T., Bovchaliuk, V., Derimian, Y., Augustin, P., Fourmentin, M., Tanre, D., Korenskiy, M., Whiteman, D. N., Diallo, A., Ndiaye, T., Kolgotin, A., and Dubovik, O.: Retrieval of optical and physical properties of African dust from multiwavelength Raman lidar measurements during the SHADOW campaign in Senegal, Atmos. Chem. Phys., 16, 7013-7028, doi:10.5194/acp-16-7013-
- 15 2016, 2016.

5

- Wandinger, U., Müller, D., Böckmann, C., Althausen, D., Matthias, V., Bösenberg, J, Weiß, V., Fiebig, M., Wendisch, M., Stohl, A., and Ansmann. A.: Optical and microphysical characterization of biomass-burning and industrial-pollution aerosols from multiwavelength lidar and aircraft measurements, J. Geophys. Res., 107(D21), doi:10.1029/2000JD000202,2002.
- Whiteman, D. N.: Examination of the traditional Raman lidar technique. I. Evaluating the temperature-dependent lidar equations, Appl. Opt.,
  42, 2571–2592, doi:10.1364/AO.42.002571, 2003a.
  - Whiteman, D. N.: Examination of the traditional Raman lidar technique. II. Evaluating the ratios for water vapor and aerosols, Appl. Opt., 42, 2593–2608, doi:10.1364/AO.42.002593, 2003b.

**Figure 1.** Sketch of the receiver unit of the lidar BERTHA with 13 detection channels (all operated in the photon counting mode): three elastic backscatter channels (355, 532 and 1064 nm), two cross-polarized elastic backscatter channels (355p, 532p, index p for perpendicular), two nitrogen vibrational-rotational Raman channels (387 and 607 nm), three rotational Raman channels (532rr, grating monochromator technique) (Arshinov et al., 2005), one rotational Raman channel (1058), a water-vapor vibrational-rotational Raman channel at 407 nm and a HSRL channel at 532 nm. All detectors are photomultiplier tubes (PMT) from Hamamatsu (H10721P-110, except for 1058 and 1064 nm). For the 1058 and 1064 nm channels the PMTs R3236 from Hamamatsu are used. To reduce the signal noise, they are cooled down to lower than -30°C.